# Explaining why increases in generic use outpace decreases in brand name medicine use in multisource markets and the role of regulation

**Katharina E. Blankart**[1,2]*, **Sotiris Vandoros**[3,4]

**1** Institute of Health Economics and Health Policy, Bern University of Applied Sciences, Bern, Switzerland,
**2** CINCH Health Economics Research Center and Faculty of Business Administration and Economics,
University of Duisburg-Essen, Essen, Germany, **3** University College London, London, United Kingdom,
**4** Harvard T.H. Chan School of Public Health, Boston, Massachusetts, United States of America

* katharina.blankart@uni-due.de

**Data Availability Statement:** There is a third-party restriction against sharing the minimal dataset imposed by the data provider of the Medimed data set. The reason is to protect confidentiality of

## Abstract

### Background

Healthcare systems worldwide face escalating pharmaceutical expenditures despite interventions targeting pricing and generic substitution. Existing studies often overlook unwarranted volume increases in multisource markets due to differential physician perceptions of brand name and generics.

### Objective

This study aims to explain the outpacing of generic medicine use over brand name use in multisource markets and assess the regulatory role, specifically examining the impact of reference pricing on volume and intensity increases.

### Methods

Analyzing German multisource prescription medicine markets from 2011 to 2014, we evaluate regulatory mechanisms and explore whether brand name and generic medicines constitute separate market segments. Using an Oaxaca-Blinder decomposition approach, we divide the differential in brand name versus generic medicine use rates into market structure and unobserved segment effects.

### Results

Generic use rates surpass same-market brand name substitution by 3.87 prescriptions per physician and medicine, on average. Reference pricing mitigated volume increase, treatment intensity and expenditure. Disparities in quantity and expenditure dynamics between brand name and generic segments are partially explained by market structure and segment effects.

physicians and patients included in the prescriber panel. A non-author point of contact that is able to receive queries regarding data access is service@medimed.info or phone +49 6251 8484-522. All data that are not subject to confidentiality agreements, program code used to generate and anlayse the data, and a data availability and provenance statement are provided via: https://osf.io/euqs9/?view_only=523df531cc8541db9ca2c7d7ed4b334e. In addition to that, we made the link to the reproduction package publicly available that documents all data: https://osf.io/euqs9/.

**Funding:** The author(s) received no specific funding for this work.

**Competing interests:** The authors have declared that no competing interests exist.

## Conclusion

Generic medicine use effectively reduces expenditures but contributes to increased net prescription rates. Reference pricing may control medicine use, but divergent physician perceptions of brand name and generics, revealed by identified segment effects, call for nuanced policy interventions.

## 1. Introduction

Increasing use of generic medicines relative to their brand-name counterparts has been used as a way to contain pharmaceutical expenditure, as they contain the same active ingredient as the corresponding originator. In Germany, generic prescriptions make up 76.4% of total expenditure in 2015, having steadily increased from 43.3% in 1996 [1]. Nevertheless, the rise in generic use has not prevented medicine utilisation from remaining constant or from increasing. It appears that expenditure increases are only partly offset by generic uptake and more intensive use of medicines, even long after patent expiry. The dynamics in medicine use have been acknowledged as a main driver of increases in health care expenditure [2]. While total prescription medicine use in Germany has increased from 28 billion to 40 billion defined daily doses (about 42 percent between 2005 and 2015), use of generic medicines has almost doubled from 16 billion to 33 billion doses in parallel leading to expenditures of 12.37 billion euros considering gross prices (or, 11.34 billion euros when excluding mandatory manufacturer and pharmacy rebates) in 2015 [1,3]. Countries like the United States have observed quantity increases at similar scale [4,5].

In this study, we aim to explain the dynamics in prescription medicine use at physician level by brand name compared to generic market segments in multisource markets of prescription medicines. Although generics are typically considered perfect substitutes to their brand name counterparts from a therapeutic perspective, brand name and generics may often be considered two distinct segments of the same market, due to the different timing in market entry owed to patent protection and pricing, as well as brand loyalty. We document and evaluate why increases in generic use outpace decreases in brand name medicine use in multisource markets. We focus on the German market and consider 45 high volume prescription medicines by active ingredient where generic competition is established and physicians can freely choose between brand name and generic medicines, although regulation typically requires them to prescribe by molecule name. We investigate the extent to which rules such as reference pricing, competition and disease prevalence, can explain differences in use of brand name and generic medicines across time. While there are additional regulations such as automatic substitution by pharmacies, tenders or influence of regulation by physicians' associations, we concentrate on elements of market structure that vary at active ingredient level to influence physician prescription. There may be differences in perceptions of brand name and generic medicines that lead to higher consumption of generics as these are typically low cost and face few prescribing restrictions after patent expiry. In that context, we evaluate the role of internal reference pricing as a policy that restricts the choice set of products in favor of lower-priced generic versions of an active ingredient to mitigate medicine volume increases.

Previous studies have quantified the role of changes in prices, quantities, technological progress or substitution effects in the evolution of pharmaceutical expenditure and pharmaceutical price inflation [5–7]. We expand this literature by considering physician prescribing to uncover the extent to which the dynamics in use of medicines across segments is driven by

physician perceptions and unobserved factors compared to market structure. We can provide insight in the extent to which the accelerated uptake of generic medicines is caused by differences in how providers use generic and brand name segments that market structure cannot explain. As opposed to previous studies that have focused on the US and market level decomposition of pharmaceutical expenditure, we consider Germany, the largest market in Europe. We then decompose the differential of the market dynamics of using brand name and generic medicines between 2011 and 2014 in a counterfactual manner to quantify the role of both market structure (i.e. observed characteristics of branded and generic markets) and segment effects (i.e. degree to which drivers in branded and generic markets differ).

Generic and brand name medicines, although identical, are often perceived as two different segments that may be attributed to differences in loyalty and persistence with prescribing brand name versus generic medicines [8]. Eliciting segment effects may explain why the increase in generic medicine use rates, that means the number of prescriptions and the resulting pharmaceutical expenditure, outpaces the decrease in brand name medicines. Varying perceptions of certain segments of goods have been acknowledged for a long time [9]. Especially goods that are perceived to contribute to a low portion of total expenditure, in our case the total use of generic versions of medicines, may be perceived differently compared to expensive items, for example expenditure for newly licensed high-cost innovative medicines.

In considering the role of internal reference pricing to drive quantities across medicines, we evaluate the long-term effects of how a policy that implicitly restricts the choice set of medicines has on quantities prescribed for generic and brand name medicines. Physicians may prescribe whatever they see appropriate, but may use medicines regulated under reference prices differently when patients may experience substitution or additional co-payments for these medicines when acting altruistically [10]. In addition, physicians have been shown to be sticky to certain prescription patterns that may differ when a choice set is regulated compared to a free choice of options [11].

Various policies to encourage generic use have contributed to effectively containing some of the expected increases in expenditures by promoting generic substitution, or prices of medicines in multisource markets, but seldom quantities. According to the US Food and Drug Administration, multisource drugs are defined where at least one other drug is available in the market that is therapeutically equivalent. Exchanging brand name for generic medicines is considered a key leading policy in curbing pharmaceutical expenditure while utilisation rates of medicines are typically not monitored as closely. The effect of reference price status of a medicine on quantity changes in the decomposition of expenditures has largely been ignored. Previous studies have concentrated on analyzing generic shares and price dynamics after patent expiry, and the short-term effects of cost-containment strategies such as reference pricing on prices, competition, cost-sharing and market shares of reference prices medicines [12–14].

Besides considering rules that restrict the choice set of physicians, we set out to study physician level heterogeneity in adopting generics over time. We consider that the use of generics would be higher would physicians immediately adopt a generic medicine instead of brand name medicines once generic versions become available. We account for the physician practice that is subject to substantial heterogeneity in adopting brand name and generic medicines. Uptake of medicines takes considerable time, and not all physicians immediately write prescriptions for an active ingredient [15–17]. Physicians typically do not adopt a medicine randomly, but choices are informed by physician and patient characteristics, learning, uncertainty of effectiveness and, network effects. [17–20]. Across segments of multisource markets, we account for that some physicians choose to adopt generic medicines later than others [21]. Physicians' generic substitution rates further depend on physician-specific factors such their mode of employment (public vs. private) [22].

## 2. Background

### 2.1 Controlling pharmaceutical expenditure in Germany

We study the dynamics of medicine use rates in the German pharmaceutical market for prescription medicines that are regulated at federal and regional level by considering prescription volume and pharmaceutical expenditure. In international comparison, Germany has high generic use rates, but overall high level of expenditure and prescription medicine use in terms of volume (538 defined daily doses per capita in 2012) [23]. Policy makers have implemented regulations to control expenditure, targeting certain medicines or segments most importantly through internal reference pricing and price negotiations for newly licensed medicines. Another approach is aiming at controlling physicians' prescription medicine expenditure and efficient use of high-volume therapeutic classes.

In ambulatory care, only physicians can issue prescriptions to patients. If there is a medical reason, physicians may rule out substitution of the medicine presentation indicated in the pharmacy (aut-idem prescribing). When patients fill their prescriptions, pharmacists are responsible for finding the cost-efficient option among the available medicines of an active ingredient in face of internal reference pricing, presence of preferred supplier contracts and parallel-trade. Physicians see list prices of medicines, but are not aware of the final price and the copayment only to a limited extent. Sickness funds have the option to negotiate preferred supplier contracts (tendering) with pharmaceutical manufacturers at the active ingredient level.

Given the regulatory environment, we assume that changes in use rates and quantity are at physician discretion, while it is pharmacists who identify the least costly option to dispense [24]. No limitations are in place to access medicines available in the market in Germany, unlike pharmacy utilization management programs active in the United States to control expenditures [25]. In addition, prescription budgets (*German*: *Richtgrößen*) and minimum use rates of certain drug classes are monitored at physician level, but this does not apply to access or volume [26,27]. These policies might thus provide incentives to prescribe more when generics become available as physicians are encouraged to prescribe generics to meet budget and quota targets.

### 2.2 Internal reference pricing

We evaluate the role of internal reference pricing to influence use rates of medicines by physicians. About 34% of medicines, 80% of prescriptions, and 33% of expenditure were covered by reference pricing in 2017 [28]. Reference pricing categorizes medicines with similar treatment effects but heterogeneous prices into one reference price group. There is substantial cost sharing of products prescribed beyond the reference price that patients need to bear [12,29]. Evidence reviews suggest that reference pricing is effective in controlling expenditure and prices in the short term [12]. Internal reference pricing can generate savings, but may also lead to an increase in prescription medicine use [30,31]. Generic substitution policies may not always be associated with savings [32], and reference pricing may lead to substitution of low-cost medicines for more costly alternatives [33].

We consider that reference pricing implicitly restricts the choice set as patients need to co-pay the price of medicines above the reference price. Products at or below the reference price within the same therapeutic class are then more attractive. Physicians that consider patient objectives and aim to minimize cost sharing will account for this financial incentive in their prescription choice [22]. At therapeutic level, there are three types of reference price categories depending on the scope of equivalent active ingredients considered: medicines (brand name and generic products) of the same active ingredient, medicines of the same pharmacological

and therapeutic class and, medicines of the same therapeutic class, but with different mode of action [34]. In this study, we concentrate on whether an active ingredient was assigned to reference pricing and we focus on the long-term consequences of reference price status on dynamics of pharmaceutical expenditures such that we do not study effects of newly assigning a reference price to an active ingredient.

We will not evaluate generic substitution as a rule that applies to all multisource medicines. Generic substitution is generally mandated where possible unless physicians explicitly exclude substitution in the pharmacy for any reason. Physicians and pharmacists are imposed to prescribe and dispense generics but this does not restrict the choice set of medicines through financial incentives by increasing co-payments. While generic use is strongly encouraged, part of the savings made by generic substitution may be offset by switching to more costly on-patent brand name active ingredients after patent expiry [35] or increased use rates driven by the same policies. In addition, there are legally imposed rebates and price freezes that equally apply to all prescriptions written that we do not consider. However, not all medicines are subject to a rebate as the rebate for generics can be lowered by lowering the price. No generic rebate applies for drugs priced 30% below the internal reference price. It is also worth noting that none of the regulations considered changed their rules during our observation period.

## 3. Methods

### 3.1 Data collection and sampling of multisource markets

We combined data from a number of sources including pharmaceutical market reports, a physician level prescription panel, pharmaceutical detailing and regulatory information (data sources and their use in the study are described in the data availability statement). Our main data source is the CEGEDIM MEDIMED panel, which is a representative panel of prescribing physicians that allows observing the dynamics in medicine use by active ingredient and physician, 2011–2014. Our level of observation are medicines by active ingredient (lowest level 5) of the Anatomical Therapeutic Chemical (ATC) Classification System, and the physician level.

As a first step, we constructed a cross-sectional dataset of a basket of 423 active ingredients that we classified by their first year of approval in Germany between 1996 and 2014. The purpose was to describe the composition of pharmaceutical expenditure in the period 2011–2014 by market age at active ingredient level. The goal was to evaluate which medicines by market age are driving pharmaceutical expenditure. The active ingredients included represent expenditures of 801 Million Euros, 13 Million prescriptions or, 10% of medicines prescribed within the prescription medicine panel.

To identify differences in changes of prescription medicine use by multisource market segment at physician level and perform econometric analyses, we constructed a second cross-sectional data set that we mapped to prescribing behavior, promotional activity and additional regulatory information. We compared generic and brand name versions of 45 active ingredients where generic entry has occurred between 2006 and 2009. That way, we ensure a homogenous basket of multisource markets where generic competition is established and internal reference prices were concluded (S1 Table). We classified products by ATC and generic or brand name segment [36]. We focused on large-volume small-molecule medicines of multisource markets with more than 2,000 prescriptions written to patients in the prescriber panel. We excluded orphan drugs, biosimilars and low-volume specialty pharmaceuticals. We ensured that markets included off-patent medicines with at least one competitor and that patent expiration does not occur in our study period. We classified generic and brand name segments by manufacturer status, and captured products provided through distributors from parallel-trade within the brand name segment. Assignment to a reference price group by active

ingredient or groups of active ingredients was based on the regulatory classification based on the status at the beginning of our observation period, that means January 2011. The active ingredients included represent expenditures of 353 Mio. Euros, 9.6 Mio prescriptions or, 13% of medicines in the prescriber panel.

We mapped the data of the 45 active ingredients to prescription data with prescription information to capture quarterly prescription volume and expenditure by physician. We focus on prescribed instead of dispensed medicines to capture physician responses to market structure and segment effects and, the resulting changes in prescription volume and expenditure. In considering the physician level, we capture expenditure levels before any substitution takes place at the pharmacy and physician responses to medicine prices as listed in prescription software. By active ingredient, we consider physicians from specialization groups to regularly prescribe that active ingredient if the total prescription volume by specialization contributes at least five percent of prescriptions within the study period. Pharmaceutical expenditure was defined as prescriptions*price per prescription. Prices are net prices as listed by manufacturers and exclude taxes (most importantly value added tax), discounts and rebates (German: Herstellerabgabepreis). We did not deduct co-payments by patients. As prices are reported monthly in our data, price changes of listed prices are considered.

By active ingredient, we calculated growth rates by outcome studied and applied the method by Bundorf [6] to decompose how much of the changes in pharmaceutical expenditure can be attributed to quantity and price changes to clarify how much of the changes in expenditure are related to volume changes. In our baseline model of the partial regressions and the Oaxaca-Blinder decomposition, we evaluate the dynamics by calculating the first differences between Q1/2011 as period 1 and Q1/2014 as period 2. To assess how the utilization differential evolves over time and the stability of market and segment effects, we generate separate decomposition estimates by varying period 2 starting from Q2/2011 up to Q1/2014. Our final analysis sample of the baseline comparison includes 108,757 physician by active ingredient observations from 2,858 physicians.

Processing of data and compilation of the analysis data set was performed using SAS software (SAS Enterprise Guide 7.15 HF9, version 9.04.01M3P062415, WX64_WKS) Copyright © 2017 SAS Institute Inc., Cary, NC, USA).

## 3.2 Physician level expenditure and prescription medicine use models

We specify a prescription medicine use model at the level of physician $i$ for active ingredient $j$ in medicine class $c$ to analyze the role of market structure including reference price status, competition and other indicators of in the dynamics of prescription medicine use over time. We consider that physicians see brand name and generic medicines as two different segments (treatments). Qualitatively, these medicines are considered identical, as they have the same active ingredient. The physician's choice of a medicine from the generic or the brand name segment can be considered as manipulable action that policy makers could influence. Market structure components determine a physician's use rate of a medicine and may bias the influence of reference price status on pharmaceutical use. We estimate separate models by market segment where $g$ is an indicator variable with $g = A$ if the medicine is a brand name medicine and $g = B$ if the medicine is a generic.

$$Y_{gij} = \alpha \cdot \text{ reference price}_j + \sum_{k=1}^{K} X_{ijk}\beta_{gjk} + v_c + u_{gij}$$
$$g = A, B$$

(1)

Variable $Y_{gij}$ are the outcomes that reflect the dynamics in prescription medicine use. The

variable *reference price$_j$* indicates whether a medicine was subject to reference price regulation or not such. Estimates of $\alpha$ reflect the contribution of reference price status in the dynamics of prescription medicine use, accounting for other market structure components reflected by the vector of covariates, **X** is the vector of covariates $X_i = [X_{ij1},\ldots,X_{ijK}]$. $v_c$ is a fixed effect for therapeutic class by ATC classification level 3. The error term $u_{gij}$ is conditionally independent of $X$, that means $E(u|X_i) = 0$.

To capture the dynamics in prescription medicine use between baseline period 1 and period 2, the four outcome variables we specify reflect the changes in quantity by number of prescriptions, number of prescriptions written to publicly insured patients (that means statutory health insurance covering about 90 percent of the German population), treatment intensity by prescriptions per patient and, pharmaceutical expenditure:

$$\boldsymbol{y_{gi}} = \Delta\gamma_{gi} = \gamma_{gi}^2 - \gamma_{gi}^1 \tag{2}$$

We assume that the decision to use a brand name or generic medicine in one period but not the other may not be a random choice and subject to sample selection. To account for adoption bias by physicians, we used selectivity adjusted regression models [37]. If physicians do not prescribe the active ingredient by segment in period 1, we assume $\gamma_{gi}^1 = 0$. According to Eq (3), we model that physicians may not have used a medicine in period 1, given that physicians may not have used the medicine in period 1:

$$\sum\nolimits_{l=1}^{L} \boldsymbol{Z_{ijk}}\gamma_{gjk} + v_c + \boldsymbol{\varepsilon}_{gij} > 0 \tag{3}$$

where $u_{gij}$ and $\varepsilon_{gij}$ have correlation $\rho_{gij}$, $\theta_{gij} = \rho_{gij}\sigma_{u_{gij}}\phi\left(\boldsymbol{Z_{ijk}}\gamma_{gjk}\right)/\Phi\left(\boldsymbol{Z_{ijk}}\gamma_{gjk}\right)$,, and $\phi$ is the standard normal density function. $\sum_{l=1}^{L} \boldsymbol{Z_{ijk}}\gamma_{gjk}$ is the vector of covariates assuming that the likelihood of adoption is a function of time since market entry, physician and practice characteristics. We disregard physicians that do not adopt a medicine in both periods such that we can only observe changes in use rates for adopting physicians. We used White standard errors to account for heteroscedasticity of the error terms.

## 3.3 Adoption bias adjusted decomposition of prescription medicine use rates by multisource market segments

To study differences in changes in use rates by generics and brand name segments between two periods, we decompose the differential based on linear regression models in a counterfactual manner (S1 File). We denote brand name medicines as market segment A and generics as market segment B. The differential in prescription medicines use rates $\left(\overline{Y}_B - \overline{Y}_A\right)$ captures the extent to which the increase/decrease in brand name medicine utilization is different compared to the increase/decrease in generic utilization. For example, if brand name utilization increases by two prescriptions and generic utilization increases by two prescriptions, the utilization differential would be zero. We use the Blinder-Oaxaca decomposition, accounting for adoption bias [38,39]. The differential between changes in medicine use rates between group *B* and *A* was computed based on Reimers [40] such that the decomposition is performed on the

selectivity corrected differential in prescription medicine use given by:

$$\hat{\Delta}_O^{\mu} = \left(\overline{Y}_B - \overline{Y}_A\right) - \left(\hat{\theta}_B \hat{\lambda}_B - \hat{\theta}_A \hat{\lambda}_A\right) = \underbrace{\left(\hat{\beta}_{Bo} - \hat{\beta}_{Ao}\right) + \sum_{k=1}^{K} \overline{X}_{Bk}\left(\hat{\beta}_{Bjk} - \hat{\beta}_{Ajk}\right)}$$

Segment effects :
degree to which drivers in generic and brand name markets differ

$$+ \underbrace{\sum_{k=1}^{K} \left(\overline{X}_{Bjk} - \overline{X}_{Ajk}\right)\hat{\beta}_{Ajk}}$$

Market structure :

component attributable to differences in the observed (4)

characteristics of generic and branded drug markets

In this approach, $\overline{Y}_{A,B}$ are the average observed levels of our outcomes for groups A (brand name medicines) and B (generic medicines). The term $\left(\hat{\theta}_B \hat{\lambda}_B - \hat{\theta}_A \hat{\lambda}_A\right)$ corrects for adoption bias to net out the estimated differences in conditional means due to selectivity of adoption at physician level. The decomposed differential consists of two parts: The first part $\left(\hat{\beta}_{Bo} - \hat{\beta}_{Ao}\right) + \sum_{k=1}^{K} \overline{X}_{BK}\left(\hat{\beta}_{Bjk} - \hat{\beta}_{Ajk}\right)$ is the unexplained effect and relates to the residual part that cannot be accounted for by differences in the determinants of medicine use. In our context, this term reflects the 'segment effects' which describe the degree to which the differential in prescription medicine use rates differs by market segment. Differences in the differential may be driven by physicians perceiving brand name and generic versions of the same active ingredient as two different market segments as suggested in studies analyzing price differentials [8].

The second part, $\sum_{k=1}^{K} \left(\overline{X}_{Bjk} - \overline{X}_{Ajk}\right)\hat{\beta}_{Ajk}$ refers to the part that is explained by group differences in the explanatory variables. We relate this explained part to market structure, which may differ by brand name and generic market segment. Market structure reflects the degree to which physicians similarly respond to market characteristics, regardless of whether the medicine prescribed is a generic or brand name medicine. We allow for an interaction term that measures the simultaneous effect of differences in segment effects and market structure.

In the prescription medicine use model (Eq (1)), we cannot manipulate the degree to which single market structure variables influence brand name or generic medicine use in a counterfactual manner. The decomposition exercise instead can identify the causal effect of prescribing a medicine from a generic compared to a brand name multisource segment on changes in differential of prescription medicine use over time [39]. The differential $\hat{\Delta}_O^{\mu}$ identifies the treatment effect to what extent generic use rates would be changing, had we considered the parameters of brand name medicines. Allowing for that brand name prescriptions are replaced by generic prescriptions, an alternative approach is to consider the net utilization differential $\left(\overline{Y}_B - \left(-\overline{Y}_A\right)\right)$. We provide decomposition estimates of the net utilization differential in S3 Table.

## 3.4 Market structure variables

At the second stage of our prescription medicine use model, the vector **X** includes a set of variables that correspond with market structure and include the following elements: reference price status of a medicine, competition, promotional activity, and patient structure. We

specified continuous variables as first differences to capture the effects of changes of market structure in terms of numbers of manufacturers available and changes in the physician's behavior compared to period 1.

The specific variables that describe the elements of market structure are as follows: We captured competition as continuous variable by the number of manufacturers offering by active ingredient level and generic or brand name segment. For brand name medicines, we count parallel trade companies as separate manufacturers. Promotional activity was measured as continuous variable by the cumulated promotional expenditures per physician at active ingredient level since market entry.

To capture patient structure, per physician, we calculated the share of patients in statutory compared to private health insurance the physician writes prescriptions for as continuous variable. Privately insured patients typically pre-pay prescriptions and may be subject to a deductible conditional on their plan. The publicly insured need to provide a co-payment between 5 and 10 euros per prescription. As physicians may rule out substitution by the pharmacist of a product specified in the prescription, as continuous variable, we captured the share of prescriptions indicated as 'aut-idem' or known 'dispense as written' in other countries [41,42]. Changes in aut-idem prescribing may translate into a change in balance between segments.

By physician, we identify physicians practicing in one region (Bavaria) without any budgetary control compared to all other regions. A policy to monitor expenditure is the total budget of the physician spent for medicines compared to a predefined spending level [26]. The Bavarian physician association has stopped to budgetary controls and mandated generic prescription quotas as of 2009.

Finally, as last element of patient structure, we considered dynamics in medicine use rates that may be due to changes in medical need [5,6]. Using the complete prescription data, we captured the number of patients in a therapeutic area at physician level. We used pharmacy-based metrics to capture the number of patients by chronic condition using the ATC classification [43]. We matched the prescription-based disease prevalence to the 45 active ingredients of our sample. We assume that physicians cannot influence the number of patients with a certain condition and that physicians do not change their coding practices of prescriptions over time. Finally, we capture the share of patients older than 65 years of age receiving the active ingredient.

## 3.5 Variables to control for physician adoption bias

For the first stage that captures the physician decision to adopt a generic or brand name medicine, the vector of covariates **Z** captures physician characteristics and indicators of the physician's prescription practice. To capture product variety prescribed, we counted the total number of ATC classes. We counted the number of patients as measure of practice size. We identified whether physicians were general practitioners or specialists, whether the physician was practicing in a group or solo practice, physician sex and, the physician's age. By active ingredient, we accounted for the timing since market entry for brand name medicines and the timing since generic entry for generics, separately to consider market age.

## 3.6 Data analysis

Econometric analyses were performed using Stata 17. We first estimated the physician level expenditure and prescription medicine use models that account for adoption bias using the regress, probit and heckman command. Using the identical estimation models, we performed the Heckman-adjusted Oaxaca-Blinder decomposition using the 'oaxaca'package (as of 2022-08-31), manually estimating the Mills-ratio and the threefold decomposition [44]. Results

tables were processesd using the 'estout'and the 'outreg2'packages (as of 2022-08-31). Figures of the descriptive analyses and estimates of the regressions were compiled using R version 4.2.2 (2021-10-31) and the 'tidyverse', 'cowplot', 'ggplot2', 'lubridate', 'readxl', 'stringr', and 'scales' packages (as of 2023-04-25).

## 4. Results

### 4.1 Prescription medicine use by market age

Based on cross-sectional data of 423 active ingredients that we classified by their first year of approval in Germany, between 2011 and 2014, for medicines approved from 1996 to 2014, pharmaceutical expenditure decreased from 60 to 56 million Euros per quarter in nominal terms not adjusting for inflation (Fig 1). The composition changed in terms of quantity and expenditure by market age. Medicines with market entries 1996 to 2000 that are typically available as generic versions and subject to generic substitution by 2011 were responsible for about two thirds (2011) or one half (2014) of pharmaceutical expenditure. This group of medicines accounted for 80% of prescription medicine use and had growth rates of 16.3% in terms of prescriptions. The latest available medicines that were approved 2011–2014 had a substantial proportion of expenditure per patient (590.27 EUR in 2014) and showed average growth rates in prescription volumes at 509% as these medicines were still diffusing into the market.

Turning to how quantities and expenditure in 45 multisource markets evolved, Table 1 shows conditional means for outcome variables, market structure and physician characteristics at baseline in 2011 and the change between Q1/2011 and Q1/2014. Use rates of generics increased by on average 6.31 prescriptions per medicine and physician, while use rates for brand name versions of the same medicines decreased by 2.44 prescriptions. This means that the decrease in brand name use rates was offset by the increase in generic prescription medicine use by 3.87 prescriptions, on average. Similarly, the decrease in prescriptions per patient in brand name medicines (-0.17) was offset by the increase for generic medicines (0.39). Increases in expenditures of generic medicines were much lower than decrease of brand name medicines (113.77 EUR vs. –222.11 EUR). Although the generics market segment substantially contributes to lowering expenditures, increases in use rates of generics strongly outpace reductions in brand name use. We also decompose whether changes in expenditure at active ingredient level were related to prices or quantities according to [6]. This method reveals that in the majority of the multisource medicines, the larger proportion of changes in expenditure were due to changes in quantity (94% on average across all active ingredients, Electronic Supplementary File, S1 Table), compared to 6% due to changes in prices. Accordingly, pharmaceutical expenditure in older multisource markets is strongly driven by volume changes, and not necessarily only price changes.

### 4.2 Reference pricing and prescription medicine use rates

The partial regression model estimates suggest that reference pricing substantially contributes to changes in medicine use across the board through reduced prescription use and increases in use rates of prescriptions per patient (Tables 2 and S2 reports second stage results). Prescriptions and prescriptions written to publicly insured patients were significantly lower in both the brand name and generic medicine segment when a reference price was active. For example, when controlling for market structure, we find that the change in brand name use was lower by 2.91 prescriptions, on average, and the change in generic use was 20.19 prescriptions lower. The number of prescriptions per patient was increased by 0.13 (87% of the mean reduction) for brand name medicines, and by 0.63 for generic medicines (154% of the mean) when a medicine is assigned to a reference price group and when we control for market structure and ATC

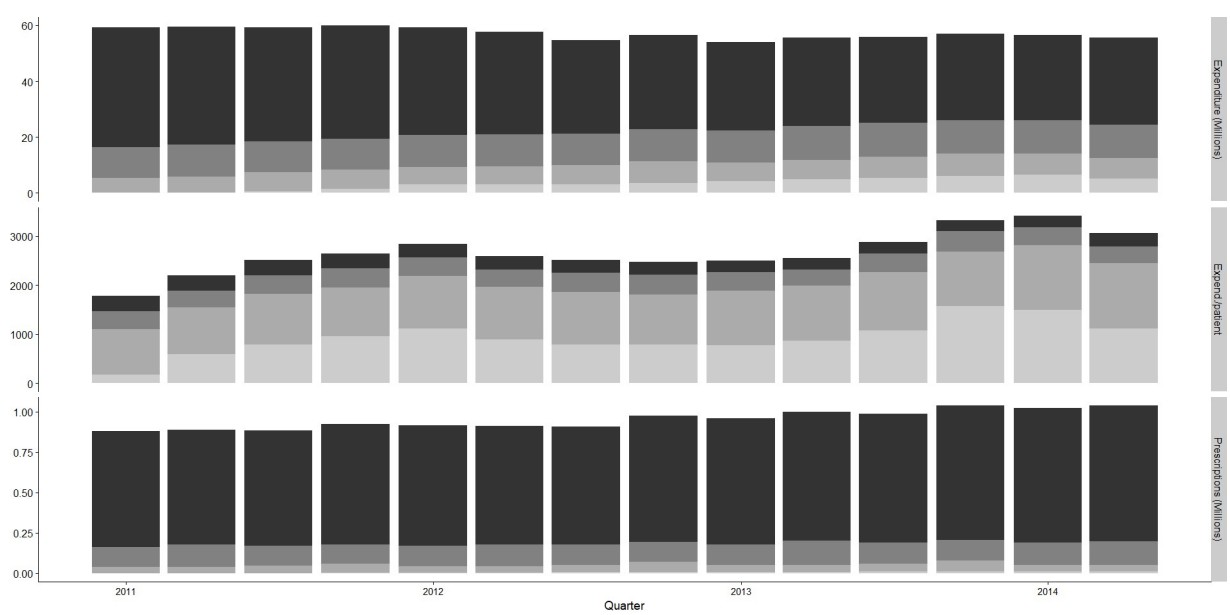

**Fig 1. Prescription volume and expenditures per patient by marketing authorization year in multisource markets.** Note: The figure shows utilization and expenditure of prescription medicines authorized between 1996 and 2014 prescribed in 2011–2014. Expenditure and prescriptions are expressed in million prescriptions. Prescription data was obtained from the CEGEDIM MEDIMED panel, 2011–2014. Data of first availability of a medicine in Germany by active ingredient was obtained from Arzneiverordnungsreport, 1997–2015. Prescription medicines approved earlier are excluded.

class fixed effects. Estimates of reference pricing status are larger when we do not control for additional variables of market structure. This means that market structure biases the impact of reference price status on medicine use. Similarly, we find that reference pricing status significantly reduced expenditures per active ingredient in brand name markets (–326.35), but was associated with increases and generic markets (936.56) when we control for market structure. The estimate for brand name medicines was positive but not significant when disregarding market structure.

The first stage results of the partial regression estimates suggest that adoption behavior by physicians over time substantially influences changes in use rates which is expressed the coefficient lambda that describes the degree of selectivity of choosing to use a brand name or generic medicine in period 2 but not in period 1. Considering adoption behavior, we find that the change in generic use is substantially higher (lambda: 6.125 prescriptions) as some physicians who have not adopted generics by period 1 just adopt by period 2. In contrast, as some physicians do not use brand name medicines in period 1 but still use them in period 2, the first stage considers that the effective decrease in use rates of brand name medicines are lower given that using brand name medicines in one period but not the other is not a random choice but driven by characteristics related to adoption behavior (lambda: -10.86).

## 4.3 Decomposition of the prescription medicine use rate differential

Table 3 reports the decomposition of dynamics in medicine use rates by market segment between the Q1/2011 and Q1/2014. The use rate differential is substantially larger when we account for adoption bias compared to the unadjusted utilization differential (–9.5 compared to –12.98 for prescriptions and –8.62 compared to –11.99 for prescriptions to publicly insured patients). The size of the differential is smaller for the differential in the number of

**Table 1. Conditional means at baseline and changes between first quarters of 2011 and 2014 in multisource markets.**

| | Brand | | | | | Generic | | | | | Mean difference |
|---|---|---|---|---|---|---|---|---|---|---|---|
| | N | mean | sd | min | max | N | mean | sd | min | max | |
| **Outcomes** | | | | | | | | | | | |
| Δ prescriptions | 56,938 | -2.44 | 9.22 | -326 | 170 | 51,819 | 6.31 | 19.05 | -270 | 458 | -8.76*** |
| | | | | | | | | | | | (-94.97) |
| Δ prescriptions: SHI | 56,938 | -2.25 | 8.78 | -322 | 134 | 51,819 | 5.71 | 17.73 | -266 | 456 | -7.96*** |
| | | | | | | | | | | | (-92.40) |
| Δ prescriptions / patient | 56,938 | -0.17 | 0.66 | -8 | 29.33 | 51,819 | 0.39 | 0.600 | -16 | 10 | -0.56*** |
| | | | | | | | | | | | (-145.86) |
| Δ pharmaceutical expenditure | 56,938 | -222.11 | 1015.75 | -47,256.1 | 24,460.87 | 51,819 | 113.77 | 517.19 | -14,560.02 | 23,680.3 | -335.9*** |
| | | | | | | | | | | | (-69.61) |
| **Market structure variables** | | | | | | | | | | | |
| Reference pricing regulated | 56,938 | 0.56 | 0.5 | 0 | 1 | 51,819 | 0.50 | 0.50 | 0 | 1 | 0.06*** |
| | | | | | | | | | | | (20.50) |
| Not reference pricing regulated | 56,938 | 0.44 | 0.5 | 0 | 1 | 51,819 | 0.50 | 0.50 | 0 | 1 | -0.062*** |
| | | | | | | | | | | | (-20.50) |
| Δ manufacturers | 56,938 | -0.31 | 0.99 | -10 | 7 | 51,819 | 1.28 | 2.22 | -8 | 16 | -1.6*** |
| | | | | | | | | | | | (-150.46) |
| Δ promotional spending | 52,411 | 572.35 | 1487.14 | -311.08 | 44,486.92 | 48,856 | 525.052 | 1,443.55 | -311.08 | 44,486.92 | 47.30*** |
| | | | | | | | | | | | (5.13) |
| Δ share public patients | 56,938 | -0.191 | 0.53 | -1 | 1 | 51,819 | 0.340 | 0.543 | -1 | 1 | -0.53*** |
| | | | | | | | | | | | (-162.12) |
| Δ share aut idem prescriptions | 56,938 | 0.011 | 0.51 | -32.5 | 46.67 | 51,819 | 0.050 | 0.43 | -48.2 | 19.83 | -0.04*** |
| | | | | | | | | | | | (-13.52) |
| Δ patient structure | 56,938 | 9.69 | 49.98 | -573 | 988 | 51,819 | 10.519 | 51.74 | -573 | 871 | -0.83** |
| | | | | | | | | | | | (-2.69) |
| Δ share patients>65 years of age | 56,938 | -0.086 | 0.475 | -1 | 1 | 51,819 | 0.209 | 0.440 | -1 | 1 | -0.295*** |
| | | | | | | | | | | | (-106.26) |
| Physician practice outside Bavaria | 56,938 | 0.952 | 0.215 | 0 | 1 | 51,819 | 0.952 | 0.214 | 0 | 1 | -0.000152 |
| | | | | | | | | | | | (-0.12) |
| Physician practice in Bavaria | 56,938 | 0.048 | 0.215 | 0 | 1 | 51,819 | 0.048 | 0.214 | 0 | 1 | 0.000152 |
| | | | | | | | | | | | (0.12) |
| **First stage variables (adoption decision)** | | | | | | | | | | | |
| # ATC classes prescribing | 56,938 | 277.36 | 49.30 | 16 | 380 | 51,819 | 277.09 | 49.585 | 16 | 380 | 0.27 |
| | | | | | | | | | | | (0.89) |
| # patients in practice | 56,938 | 1,853.71 | 931.95 | 42.67 | 20,023.67 | 51,819 | 1,854.28 | 938.10 | 42.67 | 20,023.67 | -0.57 |
| | | | | | | | | | | | (-0.10) |
| Physician sex: female | 56,938 | 0.29 | 0.45 | 0 | 1 | 51,819 | 0.29 | 0.46 | 0 | 1 | -0.00198 |
| | | | | | | | | | | | (-0.72) |
| Physician sex: male | 56,938 | 0.71 | 0.45 | 0 | 1 | 51,819 | 0.71 | 0.46 | 0 | 1 | 0.00198 |
| | | | | | | | | | | | (0.72) |
| Physician age | 56,938 | 58.20 | 6.82 | 35 | 84 | 51,819 | 58.17 | 6.86 | 35 | 84 | 0.0362 |
| | | | | | | | | | | | (0.87) |
| Group practice: no | 56,906 | 0.76 | 0.43 | 0 | 1 | 51,786 | 0.76 | 0.43 | 0 | 1 | -0.00206 |
| | | | | | | | | | | | (-0.79) |
| Group practice: yes | 56,906 | 0.24 | 0.43 | 0 | 1 | 51,786 | 0.24 | 0.43 | 0 | 1 | 0.00206 |

*(Continued)*

**Table 1.** (Continued)

| | Brand | | | | Generic | | | | Mean difference |
|---|---|---|---|---|---|---|---|---|---|
| | N | mean | sd | min | max | N | mean | sd | min | max | |
| | | | | | | | | | | | (0.79) |
| Specialist: no | 56,938 | 0.74 | 0.44 | 0 | 1 | 51,819 | 0.74 | 0.44 | 0 | 1 | -0.00368 |
| | | | | | | | | | | | (-1.38) |
| Specialist: yes | 56,938 | 0.26 | 0.44 | 0 | 1 | 51,819 | 0.26 | 0.44 | 0 | 1 | 0.00368 |
| | | | | | | | | | | | (1.38) |
| Months since market entry | 56,938 | 13.76 | 7.06 | 2 | 31 | 51,819 | 14.63 | 6.81 | 2 | 31 | -0.87*** |
| | | | | | | | | | | | (-20.89) |
| | | | | | | | | | | Observations | 108,757 |

Notes: The table shows conditional means of outcome variables, second stage variables and first stage variables of partial regressions of brand name and generic medicines used in multisource markets. Baseline variables show means in the first quarter of 2011. Variables indicating delta show the mean differential in use rates between the first quarter of 2011 and 2014. P-values are for two-sided t-tests to test differences in means between brand name and generic medicines.
* $p<0.05$
** $p<0.01$
*** $p<0.001$
Prescription data was obtained from the CEGEDIM MEDIMED panel, 2011–2014. Abbreviations: ATC: Anatomical Therapeutic Chemical Classifcation of active ingredients; SHI: statutory health insurance system.

prescriptions per patient (–0.55 unadjusted vs. –0.49 adjusted), and larger for pharmaceutical expenditure (–359.45 EUR unadjusted vs. –364.93 EUR adjusted).

The results of the decomposition allow performing counterfactual analysis of how the generics market segment would have evolved if we had applied the same market structure and segment effects as for branded medicines. We find that differences in use rates of brand name compared to generic medicines are driven by sizable segment effects (differences in endowments according to the Oaxaca-Blinder decomposition) and market structure (differences in coefficients), but results differ by outcome. For the number of prescriptions, segment effects that are due to differences in the parameters of the prescription medicine use models by generic and brand name medicines and accounted for about the other third of the differential in prescription use rates (–6.30 prescriptions). This value expresses how much of the increase in volume is related to differences perceptions or other unobservable factors that would lead to lower generic medicine use.

The size of the segment effect shows the extent to which unobservable features of the generic market segment contribute to the increase in medicine use of generics that would not be present would physicians rely on brand name medicines instead. Regarding market structure, the generic market segment would grow at a much slower pace (that means by –8.92 fewer prescriptions) would physicians prescribe according to the coefficients of the brand name segment. Market structure relates to differences in the average market structure characteristics of brand name compared to generics. Any change in a characteristic of the multisource market (for example the number of competitors or reference price status) will then translate into an actual change in prescription medicine use as much as the 'market structure' permits. The interaction effect for use rates by prescriptions was 2.24 and thus much smaller than segment effects or market structure. We find similar results for segment effects and market structure when we only consider prescriptions written to publicly insured patients.

For changes in pharmaceutical expenditure and prescriptions per patient, we find that segment effects are more important than market structure in the decomposition of the utilization

**Table 2. Adoption bias adjusted regression results of changes in medicine use rates.**

| | Prescriptions | | | | Prescriptions: SHI | | | | Prescriptions / patient | | | | Pharmaceutical expenditure | | | |
|---|---|---|---|---|---|---|---|---|---|---|---|---|---|---|---|---|
| | Brand | Brand | Generic | Generic | Brand | Brand | Generic | Generic | Brand | Brand | Generic | Generic | Brand | Brand | Generic | Generic |
| Reference pricing | -0.07 | -2.91*** | -59.84*** | -20.19** | -0.06 | -2.94*** | -54.43*** | -17.50** | 0.27*** | 0.13*** | 0.14 | 0.63*** | 122.04* | -326.35*** | -50.19 | 986.56*** |
| | (0.50) | (0.49) | (6.59) | (6.76) | (0.48) | (0.47) | (6.59) | (6.70) | (0.03) | (0.02) | (0.16) | (0.12) | (54.44) | (51.13) | (224.32) | (212.50) |
| ATC class FEs | Yes | Yes | Yes | Yes | Yes | Yes | Yes | Yes | Yes | Yes | Yes | Yes | Yes | Yes | Yes | Yes |
| Market structure controls | No | Yes | No | Yes | No | Yes | No | Yes | No | Yes | No | Yes | No | Yes | No | Yes |
| Mean | -2.477 | -2.477 | 6.515 | 6.515 | -2.28 | -2.28 | 5.897 | 5.897 | -0.15 | -0.15 | 0.41 | 0.41 | -224.41 | -224.41 | 118.292 | 118.292 |
| N | 53,823 | 53,823 | 50,269 | 50,269 | 53,823 | 53,823 | 50,269 | 50,269 | 53,823 | 53,823 | 50,269 | 50,269 | 53,823 | 53,823 | 50,269 | 50,269 |
| Adoptions | 36,980 | 36,980 | 15,761 | 15,761 | 36,980 | 36,980 | 15,761 | 15,761 | 36,980 | 36,980 | 15,761 | 15,761 | 36,980 | 36,980 | 15,761 | 15,761 |
| Chi-Squared | 2,265.1 | 5,975.0 | 2,784.8 | 5,813.0 | 2,247.6 | 5,745.2 | 2,631.1 | 5,596.7 | 1,318.5 | 33,204.3 | 1,642.3 | 17,992.4 | 3,392.1 | 10,211.9 | 940.6 | 2,851.0 |
| Lambda | 11.42 | 9.91 | -25.86 | -22.39 | 11.26 | 10.05 | -23.36 | -20.12 | -0.79 | -0.35 | 0.04 | 0.013 | 1,236.69 | 926.07 | 122.47 | 171.38 |

Note: The table displays the estimates of the changes in volumes in prescription medicine use between 2011 and 2014 per physician. The model has been adjusted by additional confounding variables of reference pricing and a fixed effect by ATC class. Standard errors are in parentheses. Abbreviations: ATC: Anatomical Therapeutic Chemical Classification of active ingredients; FEs: Fixed Effects. Market structure is defined by the number of manufacturers, promotional spending (EUR per physician), share of SHI patients, share of aut idem prescription, patient structure, share of patients > 65 years of age, considering the changes in these variables between 2011 and 2014 and, region without budget control (Bavaria). Reference pricing classification as of 2011. SHI: statutory health insurance system.

**Table 3. Decomposition of differential in prescription medicine use rates by brand name compared to generics.**

| | Prescriptions | Prescriptions: SHI | Prescriptions / patient | Pharmaceutical expenditure |
|---|---|---|---|---|
| overall | | | | |
| Brand name | -2.56*** | -2.35*** | -0.16*** | -231.39*** |
| | (0.04) | (0.04) | (0.00) | (4.55) |
| Generics | 6.94*** | 6.27*** | 0.38*** | 128.06*** |
| | (0.09) | (0.09) | (0.00) | (2.57) |
| Differential in use rates, 2011–2014 | -9.50*** | -8.62*** | -0.55*** | -359.45*** |
| | (0.10) | (0.10) | (0.00) | (5.23) |
| adjusted | | | | |
| Brand name | -2.56*** | -2.35*** | -0.16*** | -231.39*** |
| | (0.04) | (0.04) | (0.00) | (4.55) |
| Generics | 10.42*** | 9.64*** | 0.33*** | 133.54*** |
| | (0.30) | (0.29) | (0.01) | (9.04) |
| Differential in use rates, 2011–2014 | -12.98*** | -11.99*** | -0.49*** | -364.93*** |
| | (0.31) | (0.29) | (0.01) | (10.12) |
| Segment effects | -6.30*** | -6.02*** | -0.52*** | -180.62*** |
| | (0.31) | (0.29) | (0.01) | (9.29) |
| Market structure | -8.92*** | -8.42*** | 0.16*** | 129.34*** |
| | (0.31) | (0.29) | (0.01) | (12.75) |
| Interaction - segment and market structure | 2.24*** | 2.45*** | -0.13*** | -313.65*** |
| | (0.32) | (0.30) | (0.01) | (12.41) |
| N | 95,949 | 95,949 | 95,949 | 95,949 |
| Brand | 52,411 | 52,411 | 52,411 | 52,411 |
| Generics | 43,538 | 43,538 | 43,538 | 43,538 |

Note: Results report utilization levels per physician per quarter. Group 1: Brand name medicines, Group 2: Generic medicines; Prescription data was obtained from the CEGEDIM MEDIMED panel, 2011–2014. Market structure is defined by the number of manufacturers, promotional spending (EUR per physician), share of SHI patients, share of aut idem prescription, patient structure, share of patients > 65 years of age, considering the changes in these variables between 2011 and 2014 and, reference price status and region without budget control (Bavaria). Segment effects are defined by $\left(\hat{\beta}_{Bo} - \hat{\beta}_{Ao}\right) + \sum_{k=1}^{K} \bar{X}_{Bk}\left(\hat{\beta}_{Bjk} - \hat{\beta}_{Ajk}\right)$ as defined in Eq (4). SHI: statutory health insurance system

\* p<0.05

\*\* p<0.01

\*\*\* p<0.001.

differential. While segment effects structure contributed to the differential in prescription medicine use of –0.52 prescriptions per patient and –180.62 Euro of pharmaceutical expenditure, the corresponding contribution of market structure was 0.16 prescriptions per patient and 129.34 Euro of pharmaceutical expenditure. Generic medicine use would increase by 0.16 prescriptions per patient and had higher 129.34 Euro higher pharmaceutical expenditure would we apply the coefficients of brand name medicines to generics. Segment effects would lead to a lower in treatment intensity of generic medicines and lower expenditures would we apply the brand name market characteristics to the generic medicine segment. In combination with the results for prescriptions, the results suggest that differences in perceptions between

generic and brand name medicines due to segment effects lower the gap in terms of all outcomes. For prescriptions per patient, interaction effects were moderate. For expenditures, we find substantial interaction effects between market structure and segment effects that suggests sizable inter-market segment interactions.

The relative composition of market structure and segment effects in multisource markets are manifest and mostly independent on the timing of period compared to the baseline period except for short-term comparisons (Fig 2). The absolute size of the differential increases the further away the baseline period. Panel A of Fig 2 shows the adjusted utilization differential, segment effects, market structure and interactions by outcome for each quarter between Q2 2011 to Q1 2014, using Q1 2011 as baseline quarter. Panel B shows the proportion of segment effects and market structure as percentage of the total differential, including interaction effects. The differential in prescription medicine use of the current quarter compared to the Q1 2011 increases in magnitude over time. Across all time periods, the mean proportion of the explained part (that means market structure) was 23.75 percent (s.d. 90.25) for the number of prescriptions, –3.12 percent for the number of prescriptions of publicly insured patients (s.d. 7.5), –105.12 percent (s.d. 35.41) for prescriptions per patient and, 73.5 percent (s.d. 33.00) for expenditures.

The interpretation of the role of market structure and segment effects does not alter substantially when we changed the reference category to the generic market segment or, when we consider the net utilization differential to account for that brand name medicines are typically replaced by generics (S4 Table). Magnitudes of the decomposition elements varied somewhat in size.

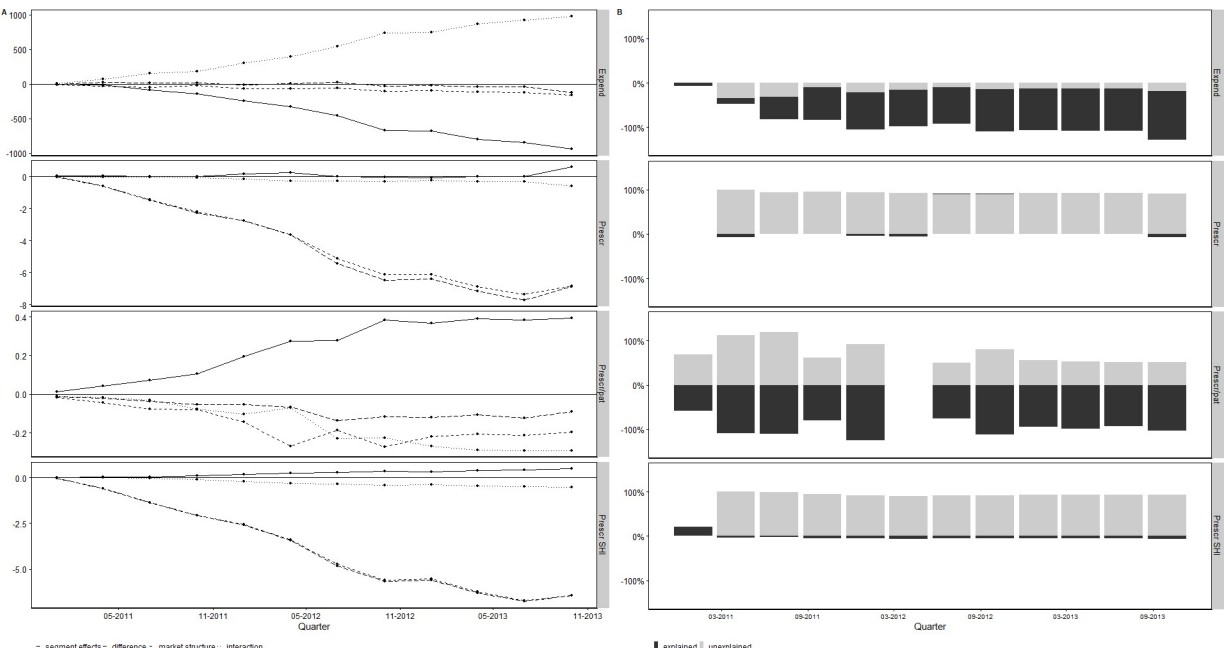

**Fig 2. Decomposition of differential in use rates of brand name and generic medicines.** Note: The figure shows the utilization differential in medicines use rates of generic and brand name medicines decomposed by market structure, segment effects, interaction between market structure and segment effects (dashed line) and the difference (solid line) (Panel A) and the proportion of the explained and unexplained part of the decomposition (Panel B) by medicine use rate related outcomes by quarter from Q1/2011 to Q1/2014. In panel B, values larger than 150% were excluded from the figure. Prescription data was obtained from the CEGEDIM MEDIMED panel, 2011–2014. Expend: expenditure; Prescr: prescriptions, pat: patient.

## 5. Discussion and conclusion

This study demonstrates how quantity increases in older multisource markets may contribute to the dynamics in pharmaceutical expenditure. This observation may be ignored by pharmaceutical policies that often target prices of new medicines without generic or biosimilar competition, or relative shares of generic compared to brand name medicines, but not access to medicines or volumes to determine the right amount of use. For the 45 multisource markets, the increase in generic use beyond the reductions in brand name use of the same market may not reflect the right amount of use despite substantial increases in generic share, which is often primary target to monitor pharmaceutical expenditure in multisource markets. The increase in generic use may stem from overuse, particularly in instances of overprescribing. Alternatively, it could be the result of previous underuse when medical needs were not adequately addressed. A third possibility could be that the increase in utilization might be a result of substituting other on-patent medicines with the newly available off-patent drug in the same class once they become available. If this is a result of the latter, these dynamics might contribute to reduced expenditure. Of course, this phenomenon might be a result of a combination of different factors.

Current pharmaceutical policy largely ignores how quantities develop over time and the role of differences in perceptions of the brand name and generic market segment. In Germany, pharmaceutical policy is focused to contain spending by continuing global price freezes of prescription pharmaceuticals and to control price setting of new medicines upon market entry. Similarly in the United States, the Inflation Reduction Act of 2022 targets selected high cost medicines with one manufacturer and no generic or biosimilar competition, which is responsible for 60% of spending [45,46]. While the 2022 law is a substantial intervention in terms of regulating prices of medicines and likely cost-effective, prices and quantities of the 40% of spending that largely comprise older multisource markets reflecting 93% of covered medicines remain uncontrolled. The substantial segment effects that this study identifies suggest that some of the volume increases in these older markets may not necessarily be substantiated by medical need and may be due to different perceptions of providers and unobserved factors of the different brand name and generic segments.

A step for further research is to analyze whether the volume increases reflect the right amount of use of an active ingredient within a therapeutic class. For example, patients may benefit from substitution of simvastatin to the somewhat more effective atorvastatin within the set of available statins in lowering lipid-levels, but not considering side effects [47–49]. If all the increases in medicine use are caused by substitution between active ingredients from lower to higher effectiveness or higher to lower prices, then any quantity effects may be welfare improving. Another explanation may be that, in the presence of cost-control measures, physicians hold back brand name treatments with the least therapeutic benefit from patients. On the contrary, if medicines are of lower quality and used as additional treatments without any additional health effects in patients with no or low marginal benefit, the segment effects we identify may require monitoring by third party payers. Therefore, after ensuring access to effective medicines, the long-term value of pharmaceutical care provided could be a subsequent goal.

Our study adds to the literature of physician decision making that deals with how physicians are driving medicine use [50]. We contribute to the literature that examines how medical technology is diffusing within health care systems due to practice variation. The sizable segment effects across all outcomes that we identify confirm that changes in medicine use are not solely due to market structure effects, but also relate to physician behavior. While we cannot disentangle whether the behavior that lead to segment effects may be sticky, persistent or

altruistic, previous studies have documented these behavioral components of prescription decisions [11,19,51]. Coscelli (19) as well as Crea et al. (11) show that habit persistence are important drivers of physician decisions in selected drug classes, but not altruism and moral hazard do not drive prescription decisions. Using strategic survey questions instead of prescription data, Cutler et al. demonstrate that physician beliefs about treatments are much more important than organizational factors or patient demand (51). Other studies have characterized early users of new medicines [17], characteristics that we consider to drive the adoption decisions of generic and brand name medicines alike.

Considering the German market, our approach complements previous evaluations that have evaluated the role of reference pricing and market segmentation. Considering the impact of reference pricing, the previous studies have demonstrated increases in health care use outside prescriptions like physician or hospital visits at the patient level [52] and identified price reductions [53]. At the physician level, we demonstrate that this policy leads to reductions in medicine use rates by prescriptions, more intensive use per patient, but net increases in expenditures at the molecule level. For market segmentation and at the consumer level, Herr and Suppliet have documented different price responses by generic and brand name products when there are price limits that lead to exemptions in co-payments in the context of reference pricing [54]. All of these studies have considered price responses, market shares, or health care use outside medicine prescribing, but not medicine use by volume and expenditure over time.

## 5.1 Limitations

Our methodological approach is subject to limitations. We used a relatively short study period, which may not allow us to capture the full dynamic effects of pharmaceutical markets. An advantage of our study period is that there was no major regulatory change interfering. Physicians who did not adopt a medicine in both periods are excluded, which might lead to a sample selection bias, as of course their choice is endogenous to such physician characteristics. Our data are limited to the number of prescriptions and prescriptions per patient instead of standard units or defined daily doses. Previous studies have shown similar effects on changes in prescription medicine use in the German market independent from how prescription volume was expressed [15]. In contrast to data from sickness funds, we derive pharmaceutical expenditure from prescription data. Accordingly, the values of pharmaceutical expenditure are higher than actually paid by statutory health insurance as we cannot account for automatic dispensing in the pharmacy to identify the cheapest option. In accounting for selectivity of prescription medicine use over time, the Heckman selection approach has been criticized for its sensitivity to model specification and distributional assumptions [55]. There may be different ways in accounting of the selectivity bias when decomposing the differential [56,57]. Finally, we cannot quantify health effects of how the increases in medicine use rates contribute to health improvements.

## 5.2 Implications for policy

The adaptation of a decomposition method allows evaluating the dynamics in pharmaceutical expenditure across time by physician level changes in pharmaceutical use rates. Our results reveal that different market structures (as captured by reference pricing, competition and other factors) and perceptions of brand name compared to generic medicines make generic uptake outpace the decrease in brand name medicines in older multisource markets. Internal reference pricing that originally intends to control prices through increased cost sharing was equally effective in both market segments to mitigate prescription intensity and expenditure, but not

quantity at the physician level. Related policies that target generic shares of physicians in selected medicine classes were effective in increasing the generic compared to brand name use.

Our results provide implications for evaluation of pharmaceutical expenditure and market design of multisource markets. As generic substitution is mandated in Germany and similarly in other countries, the sizable segment effects we uncover suggest that policies that generally encourage use of medicines from the generic segment lead to a more careful use of brand name medicines in terms of quantities, but not for generics. This observation is in line with research from other fields suggesting that consumption patterns of the same quality good may differ if the price of that good is substantially lower. Our results are also in line with that physicians seem to pay more attention to brand name medicines' use rates that may be considered big ticket items [9].

The selectivity adjusted Blinder-Oaxaca decomposition approach allows quantifying the degree to which physicians consider the markets of generic and brand name medicines of multisource markets as different market segments and to which market structure leads to different outcomes within the same active ingredient. We observe that the segment effects and the market structure effects are both sizeable and comparable. Previous approaches have identified large effects of changes in quantity of pharmaceuticals prescribed relative to changes in prices [6,58] and the potential roles of different segments on changes in overall expenditures [5]. We add to this approach by explicitly quantifying how much of the change in a certain segment is due to market structure and how much is due to segment effects. The decomposition method is not confined to comparisons of generic and brand name segments, but allows medicines by reference price status or, multisource compared to single source (on patent) markets.

## Supporting information

**S1 File. Description of counterfactual manner of Oaxaca-Blinder decomposition approach.**
(DOCX)

**S1 Table. Active ingredients of multisource markets included in decomposition analysis, prescription medicine use rates, 2011–2014.** Note: Prescription data was obtained from the CEGEDIM MEDIMED panel, 2011–2014.
(DOCX)

**S2 Table. First stage partial regression estimates of likelihood to Newly Adopt brand name or generic medicine.** Note: The table displays the estimates of the adoption model to use a medicine by segment in 2014 having the medicine not used in 2011. Standard errors are in parentheses. * p<0.05, ** p<0.01, *** p<0.001.
(DOCX)

**S3 Table. Decomposition of differential in use of brand name compared to generic medicine, net prescription medicine use.** Note: The table shows estimates of the net differential in medicine use by brand name and generic medicines at active ingredient and physician level in 45 markets, 2011–2014. Compared to the baseline specification, it is assumed that brand name prescription medicines are replaced by generic prescriptions over time after patent expiry such that the differential in prescription medicine use is defined as $\left(\overline{Y}_B - \left(-\overline{Y}_A\right)\right)$. The analysis is identical to the Oaxaca-Blinder decomposition as section 2.4. For example, if generic utilization increases by two prescriptions and brand name utilization decreases by two, the differential prescription medicine use is zero. In the original specification, the utilization differential is four. Prescription data was obtained from the CEGEDIM MEDIMED panel, 2011–2014.; * p<0.05, ** p<0.01, *** p<0.001.
(DOCX)

**S4 Table. Decomposition of differential in prescription medicine use rates by generics compared to brand name medicines.**
(DOCX)

## Acknowledgments

We are grateful to Simon Decker, Joost Enzing, Michael Stucki and Nicolas Ziebarth, and the participants of various workshops and conferences, especially the CINCH Health Economics Research Seminar, German Health Economics Association (dggö) annual meeting 2019 Augsburg, Germany, Essen Health Conference 2019, Germany, the Swiss Health Economics Workshop 2019, Lucerne, Switzerland, the IHEA Annual World Congress on Health Economics 2019, Basel, Switzerland) and the LoLa Health Economics Study Group 2020, virtual workshop, for comments.

## Author Contributions

**Conceptualization:** Katharina E. Blankart, Sotiris Vandoros.

**Data curation:** Katharina E. Blankart.

**Formal analysis:** Katharina E. Blankart.

**Methodology:** Katharina E. Blankart, Sotiris Vandoros.

**Project administration:** Katharina E. Blankart.

**Software:** Katharina E. Blankart.

**Validation:** Katharina E. Blankart, Sotiris Vandoros.

**Visualization:** Katharina E. Blankart.

**Writing – original draft:** Katharina E. Blankart.

**Writing – review & editing:** Katharina E. Blankart, Sotiris Vandoros.

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
