## [Decision Letter · Decision Letter 0]

13 Jun 2023

PONE-D-23-12926Explaining why increases in generic use outpace decreases in brand name medicine use in multisource markets and the role of regulationPLOS ONE

Dear Dr. Blankart,

Thank you for submitting your manuscript to PLOS ONE. After careful consideration, we feel that it has merit but does not fully meet PLOS ONE’s publication criteria as it currently stands. Therefore, we invite you to submit a revised version of the manuscript that addresses the points raised during the review process.

We look forward to receiving your revised manuscript.

Kind regards,

Dzintars Gotham

Academic Editor

PLOS ONE

Additional Editor Comments:

Thank you for this very interesting study. The peer review comments appear interesting and constructive.

I have only one comment to add from my side:

The authors demonstrate that the increase in generic use is greater (for many molecules) than the decrease in use of the respective brand name medicine.

Throughout the paper, including nearly all of the Discussion, the implicit argument given is that the most likely explanation is inappropriate overuse of the generic.

In lines 466-468, the authors do note that it is possible that this same observation would be seen if there was inappropriate UNDERuse of the brandname prior to generic entry. Or to put it in simple terms, that for some patients, the expensive medicine was not received even when it would have been clinically appropriate, and now with lower prices more patients are receiving it (appropriately).

But it would seem that this potential explanation - of UNDERuse of brandname prior to generic entry - is at least as possible as the explanation given priority by the authors - that there is inappropriate OVERuse of the generic.

Please add discussion of why the data demonstrates that one of these explanations is more likely than the other. Or, if it is impossible to say which is more likely based on available data, please highlight these two as equally possible explanations.

Reviewers' comments:

Reviewer's Responses to Questions

**Comments to the Author**

1. Is the manuscript technically sound, and do the data support the conclusions?

Reviewer #1: Partly

Reviewer #2: Partly

2. Has the statistical analysis been performed appropriately and rigorously? 

Reviewer #1: Yes

Reviewer #2: I Don't Know

3. Have the authors made all data underlying the findings in their manuscript fully available?

Reviewer #1: No

Reviewer #2: Yes

4. Is the manuscript presented in an intelligible fashion and written in standard English?

Reviewer #1: Yes

Reviewer #2: Yes

5. Review Comments to the Author

Reviewer #1: Summary

The paper analyses an important topic on effects of drug regulation in Germany and contains interesting results on market design. It uses adequate data and statistical analysis. Overall, the paper is not always easy to follow. Variables should be defined and described more clearly and more explanations and discussions seem useful.

Major Issues

Page 8, line 160: It appears that prices of drugs are not explicitly taken into account, but implicitly based on the induced expenditure . Is this correct? If data on expenditure come from prescriptions, you might not consider substitution which bias the real expenditure. Please explain in more detail.

Page 8. Line 164: Which price level is used to calculate the expenditure (a) manufacturer or pharmacy selling price, b) gross price or net price, c) if net price: what is deducted: legally defined rebates, patient co-payment; discounts from tenders are not deducted)? What do you mean with “before taxes”? Is this the price without value added tax?

Page 8, line 164: Have prices changes over time have been taken into account?

Chapter 3.3.: The terms “adoption bias” and “market segment” are not precisely defined. How are the variable defining adoption bias and market segments coded?

Chapter 3.4. The term “market structure” also used in Table 2 , for instance, should be defined more precisely. Which variable define which kind of market structure? How are those coded?

Page 13, lines 264-265: You use the share of privately insured patients to control for this factor. However, privately insured are a different market with different regulation. IRP, automatic substitution and regulations by the physicians associations do not apply to privately insured persons. You might discuss whether controlling for privately insured persons is enough in your study. However, I would suggest to exclude privately insured patients from the analysis to reduce bias, because your research question is in IRP which does not apply to privately insured patients.

Page 18, line 361: A central result is that after adoption bias adjustment, the number of prescriptions per patient seems to be reduced by reference pricing. Table 1 shows the availability of generics which provides a cheaper alternative reduced brands and rises generic as well as overall prescriptions per patients. Why does the further impact of IRP do not work into the same directions? This appears counterintuitive. Please explain this central result in more depth in the results and/or discussions section.,

Table 4: How do you define and code the variables “market structure” and “segments effects”?

Page 23, line 459: Why is the increase in prescription volume not necessarily substantiated by medical need? How is this based on the data and analysis? One can regularly observe that the volume of prescriptions increase after generic entry and start of price competition in contrast to the monopoly of the patented brand. Economic theory suggests that the monopolistic price of the branded drugs results in higher price and lower volumes compared with the generic market structure which reflects competition. Based on this, the monopoly market structure is likely inducing undersupply not reflecting the medical need.

Page 24, line 475: specify the different market structure here. Some readers might focus n the conclusions section only.

Suppl. A6: You use the 2012 version to capture reference pricing. Why not 2011? What about drugs being assigned to a IRP group between period 1 and 2?

Minor Issues

The conclusion in the abstract is vague. In which way should access to medicines be managed?

Different types of referencing are used. Reference Albrecht el al. in the supplement is not number [50].

Page 1, line 11: Is this based on gross or net prices. Do the expenditure reflect manufacturer or pharmacy retail prices.

Page 1, line 23: Market structure is more complex, i.e. automatic substitution, tenders, influence of regulation by physicians’ associations (KV). This might be stated here.

Page 3, line 51: How is the choice set restricted by internal reference pricing (IRP)? What is the proposed mechanism? Physicians may prescribe whatever they see appropriate in Germany. In most situation, manufacturers lower their prices such that patients do not have to pay a positive difference between prices and reference price.

Page 5, lines 97: You might discuss potential incentives of regulation by physicans’ associations. Quotas as used in Bavaria or “Richtgrößen” might induce incentives to prescribe more when generics become available. Generics can lower the average cost of prescriptions per patient such that there is more room to prescribe drugs for a patient, e.g. prescribing of two generic drugs to control blood pressure instead of using a branded combination or a single branded drug.

Page 6, line 125: The rebates do apply to all drugs, but are not equal. The rebate for generics can be lowered by lowering the price and no generic rebate applies for drugs priced 30% below the internal reference price.

Page 6, line 136: Does this mean that your data reflect the specific drug prescribed by the physician and not the dispensed drug after automatic substitution?

Chapter 3.6.: Might be part of discussion after the results chapter.

Table 1: Please explain the meaning of the variable “patients with indication”.

Table 2: put “brand” and “generic” between the columns to which these refer.

Page 20, line 390; page 21, line 412/412: It seems that an “if” is missing in the senctence.

Page 20, line 392: What do you mean with “endowments”? Please define more precisely.

Reviewer #2: This paper uses a data set on prescription data from Germany to examine the utilization patterns of brand and generic drugs. The study tackles a pertinent issue and offers noteworthy insights, thereby warranting its publication. Nevertheless, the paper's contribution and the discussion of its findings would benefit from being more prominent. In the following section, I elaborate on this issue and list several minor comments.

My main criticism is that the paper lacks a clear discussion of the results in relation to the existing literature. The authors state that their contribution lies in:

• Considering physicians' prescribing behavior.

• Considering the German market (previous studies focused on the USA).

The discussion only partially touches upon these points, and there is a lack of clear comparison with the existing literature. The authors should elaborate on this.

Minor comments:

The findings section in the abstract has the potential to mislead readers. It should be highlighted that while generics are effective in reducing expenditures, they come at the cost of increased net prescription rates (provide both delta prescriptions and delta expenditures).

Bar over YA, YB seems not be introduced as notation.

P16, line 343: Where do the percentages come from?

Table 1: are these all variables the authors use? Please clarify what belongs to market structure (only the number of manufacturers?) etc. additional descriptive statistics would be helpful, i.e. upper and lower bounds.

“Estimates of reference pricing status are larger when 364 we do not control for market structure” So the number of competitors lowers the prices? Please elaborate.

Line 389: The paragraph solely presents technical arguments. What are the implications of these findings? Why would the same market structure lead to a decrease of -5.45 prescriptions?

6. PLOS authors have the option to publish the peer review history of their article (what does this mean?). If published, this will include your full peer review and any attached files.

Reviewer #1: No

Reviewer #2: No

---

## [Author Response · Author response to Decision Letter 0]

27 Dec 2023

Please refer to detailed revision note for responses to comments.

---

## [Decision Letter · Decision Letter 1]

31 Jan 2024

PONE-D-23-12926R1Explaining why increases in generic use outpace decreases in brand name medicine use in multisource markets and the role of regulationPLOS ONE

Dear Dr. Blankart,

Thank you for submitting your manuscript to PLOS ONE. After careful consideration, we feel that it has merit but does not fully meet PLOS ONE’s publication criteria as it currently stands. Therefore, we invite you to submit a revised version of the manuscript that addresses the points raised during the review process. Please submit your revised manuscript by Mar 16 2024 11:59PM. If you will need more time than this to complete your revisions, please reply to this message or contact the journal office at plosone@plos.org. Please include the following items when submitting your revised manuscript:A rebuttal letter that responds to each point raised by the academic editor and reviewer(s). You should upload this letter as a separate file labeled 'Response to Reviewers'.A marked-up copy of your manuscript that highlights changes made to the original version. You should upload this as a separate file labeled 'Revised Manuscript with Track Changes'.An unmarked version of your revised paper without tracked changes. You should upload this as a separate file labeled 'Manuscript'.If applicable, we recommend that you deposit your laboratory protocols in protocols.io to enhance the reproducibility of your results. Protocols.io assigns your protocol its own identifier (DOI) so that it can be cited independently in the future. For instructions see: https://journals.plos.org/plosone/s/submission-guidelines#loc-laboratory-protocols. Additionally, PLOS ONE offers an option for publishing peer-reviewed Lab Protocol articles, which describe protocols hosted on protocols.io. Read more information on sharing protocols at https://plos.org/protocols?utm_medium=editorial-email&utm_source=authorletters&utm_campaign=protocols.

We look forward to receiving your revised manuscript.

Kind regards,

Dzintars Gotham

Academic Editor

PLOS ONE

Journal Requirements:

Additional Editor Comments:

Thank you for these thorough revisions. A few requests remain from Reviewer #2, which I hope are relatively easy to address.

Reviewers' comments:

Reviewer's Responses to Questions

**Comments to the Author**

1. If the authors have adequately addressed your comments raised in a previous round of review and you feel that this manuscript is now acceptable for publication, you may indicate that here to bypass the “Comments to the Author” section, enter your conflict of interest statement in the “Confidential to Editor” section, and submit your "Accept" recommendation.

Reviewer #1: All comments have been addressed

Reviewer #2: (No Response)

2. Is the manuscript technically sound, and do the data support the conclusions?

Reviewer #1: Yes

Reviewer #2: Yes

3. Has the statistical analysis been performed appropriately and rigorously? 

Reviewer #1: Yes

Reviewer #2: Yes

4. Have the authors made all data underlying the findings in their manuscript fully available?

Reviewer #1: Yes

Reviewer #2: Yes

5. Is the manuscript presented in an intelligible fashion and written in standard English?

Reviewer #1: Yes

Reviewer #2: Yes

6. Review Comments to the Author

Reviewer #1: All comments have been adequately adressed. The paper gives clear insight into dynamics of the pharmaceutical market when generics enter.

Reviewer #2: The authors have addressed most of my comments appropriately. The comparison with literature is, however, still too vague and must be improved before publication. See comment 1.

Comment 1: Comparison against literature

It is positive to have an overview of the related literature now. However, the description of the papers' contributions is rather vague and challenging to follow. For instance, I examined Stargardt, reference 52, in “Considering the impact of reference pricing, the previous studies have demonstrated increases in health care use outside prescriptions at the patient level (52) and identified price reductions (53). Stargardt investigates “patients with prescriptions for statins.” To what metric/variables in (52) do you refer in “health care use outside prescriptions”?

Likewise, the usage rate has not been defined in the paper. This is a central topic in the paper and should be briefly defined to avoid any misinterpretation.

Comment 2: Table 3

The values in Table 3 have considerably changed. I assume this is due to the recalibration mentioned in the note to the editor. Please clarify.

7. PLOS authors have the option to publish the peer review history of their article (what does this mean?). If published, this will include your full peer review and any attached files.

Reviewer #1: No

Reviewer #2: No

---

## [Author Response · Author response to Decision Letter 1]

15 Mar 2024

Please refer to the detailed revision note uploaded.

---

## [Editor Report · Decision Letter 2]

20 Mar 2024

Explaining why increases in generic use outpace decreases in brand name medicine use in multisource markets and the role of regulation

PONE-D-23-12926R2

Dear Dr. Blankart,

We’re pleased to inform you that your manuscript has been judged scientifically suitable for publication and will be formally accepted for publication once it meets all outstanding technical requirements.

Kind regards,

Dzintars Gotham

Academic Editor

PLOS ONE